# Investigating Linguistic Steering: An Analysis of Adjectival Effects Across Large Language Model Architectures

**Lars Malmqvist**                                                                                   *lars@resai.dk*
*Research and Implementation*

**Reviewed on OpenReview:** *https://openreview.net/forum?id=6670*

## Abstract

Achieving reliable control of Large Language Models (LLMs) requires a precise, scalable understanding of how they interpret linguistic cues. We introduce a rigorous framework using Shapley values to quantify the observed influence of individual adjectives on model performance, moving beyond anecdotal heuristics to principled attribution. Applying this method to 100 adjectives across a diverse suite of models (including o3, gpt-4o-mini, phi-3, llama-3-70b, and deepseek-r1) on the MMLU benchmark, with a cross-benchmark validation on ARC-Challenge, we uncover several critical findings for AI alignment. First, we find that a small subset of adjectives act as disproportionately powerful "levers," yet their effects are not universal. Cross-model analysis reveals a "family effect": models of a shared lineage exhibit correlated sensitivity profiles, while architecturally distinct models react in a largely uncorrelated manner, challenging the notion of a one-size-fits-all prompting strategy. Second, focused follow-up studies demonstrate that the steering direction of these powerful adjectives is not intrinsic but is highly contingent on their syntactic role and position within the prompt. For larger models like gpt-4o-mini, we provide the first quantitative evidence of strong, non-additive interaction effects where adjectives can synergistically amplify, antagonistically dampen, or even reverse each other's impact. In contrast, smaller models like phi-3 exhibit a more literal and less compositional response. These results suggest that as models scale, their interpretation of prompts becomes more sophisticated but also less predictable, posing a significant challenge for robustly steering model behavior and highlighting the need for compositional and model-specific alignment techniques.

## 1 Introduction

While prompt engineering techniques like Chain of Thought (CoT) (Wei et al., 2022) are effective at influencing the behavior of Large Language Models (LLMs), the practice remains largely heuristic. This lack of a principled understanding of how individual linguistic elements affect model outputs presents a barrier to achieving the predictable control necessary for robust AI alignment (Christian, 2020). A systematic methodology is required to move from crafting prompts to analyzing them.

This paper addresses this gap by introducing a scalable, game-theoretic framework to deconstruct and quantify the influence of individual words within a prompt. We utilize Shapley values (Shapley, 1953), a method from cooperative game theory that provides a principled means for attributing the contribution of each feature to a model's prediction. Using the KernelSHAP approximation (Lundberg & Lee, 2017), we analyze the impact of 100 common adjectives on the performance of five distinct LLMs on the Massive Multitask Language Understanding (MMLU) benchmark (Hendrycks et al., 2021). Our analysis yields several findings regarding the patterns of linguistic influence.

First, we establish that a small subset of adjectives functions as high-impact "levers," while the majority have a negligible effect, a principle that holds across all tested architectures. Second, we identify distinct styles of interpretation. Advanced models exhibit up to an order of magnitude greater sensitivity to adjectival

modifiers than smaller open source models. Third, we find that the directional impact of these levers is not universal but is subject to a "lineage effect," with models from the same developer showing correlated sensitivity profiles that are distinct from those of other model families. Our analysis also shows that models do not share a universal hierarchy of task sensitivity. Instead, each model family exhibits a unique "domain sensitivity signature." Finally, follow-up studies demonstrate that these effects are modulated by prompt syntax and compositional word interactions, with persona-based instructions identified as a particularly influential prompting mechanism.

This work provides a detailed map of adjectival sensitivity across a range of modern LLMs. The results indicate that while the existence of influential linguistic levers is a consistent property of these systems, the specific vocabulary and context for effective influence are highly model-dependent. This suggests the necessity of an empirical, model-specific approach to developing robust alignment and control techniques using prompt engineering.

## 2 Related Work

This research is situated at the intersection of three primary areas of study: prompt engineering, model interpretability, and AI alignment. Our work builds upon methods from each of these fields to provide a new type of analysis.

### 2.1 Prompt Engineering and In-Context Learning

The capacity of LLMs to perform tasks based on a few examples provided in the prompt, known as in-context learning, was a central finding of the work on GPT-3 (Brown et al., 2020). This led to the development of prompt engineering, a practice focused on designing inputs that elicit desired behaviors. A significant advancement in this area is Chain of Thought (CoT) prompting, which instructs models to generate step-by-step reasoning before providing a final answer, thereby improving performance on complex tasks (Wei et al., 2022). Subsequent work has expanded on this by generating multiple reasoning paths and selecting the most consistent answer (Wang et al., 2023) or by exploring thoughts in a tree structure (Yao et al., 2023).

Other research has focused on automating the creation of effective prompts. Methods like AutoPrompt use gradient-based searches to find optimal discrete tokens for prompts (Shin et al., 2020). More recent work, such as Automatic Prompt Engineer (APE), uses LLMs themselves to generate and select instructions for other models (Zhou et al., 2022). Complementary to our adjective-based analysis, Miehling et al. (Miehling et al., 2025) publish the first benchmark quantifying how far a model's persona distribution can be shifted by prompting, revealing systematic asymmetries in steerability across model families.

While these studies are effective at optimizing prompts as complete units, our work differs in its objective. Instead of searching for an optimal prompt, we deconstruct existing prompts to analyze the contribution of their individual linguistic components. Our analysis is complementary, offering a method to understand why certain automatically generated prompts might be effective.

### 2.2 Interpretability and Feature Attribution in NLP

Understanding the internal workings of complex models is a long-standing goal of machine learning research. Model-agnostic techniques, which treat the model as a black box, are applicable to the closed, API-based models used in this study. Methods like LIME explain predictions by building a simple, interpretable local model around a specific input (Ribeiro et al., 2016). Our work is based on SHAP (Lundberg & Lee, 2017), which uses a game-theoretic foundation to attribute a prediction to the model's input features. Goldshmidt and Horovicz (Goldshmidt & Horovicz, 2024) propose TokenSHAP, a Monte-Carlo Shapley estimator that assigns importance to individual prompt tokens. Building on this, Amara et al. (Amara et al., 2025) introduce ConceptX, which elevates Shapley attribution to the concept level and demonstrates how such explanations can both audit and *steer* LLM outputs. While interesting neither of these approaches are directly applicable to our case.

Within NLP, other interpretability approaches examine model internals. Early work focused on visualizing attention weights, though subsequent studies have shown that attention is not always a reliable proxy for feature importance (Jain & Wallace, 2019). A more recent direction is mechanistic interpretability, which aims to reverse-engineer the specific circuits and algorithms that transformers learn by analyzing individual neuron activations and their connections (Olah et al., 2020; Bau et al., 2017). Our study occupies a middle ground. It remains model-agnostic, requiring no access to internal states, but provides a more granular analysis than simple input-output tests by treating individual words as the fundamental features whose contributions are to be measured.

### 2.3 AI Alignment and Controllability

The broader goal of this work is to contribute to the alignment of AI systems with human intent (Christian, 2020). The dominant method for achieving this has been Reinforcement Learning from Human Feedback (RLHF), which fine-tunes a pretrained model using a reward model trained on human preference data (Christiano et al., 2017; Ouyang et al., 2022). An alternative approach that avoids training a separate reward model is Direct Preference Optimization (DPO), which uses a direct classification loss on preference pairs (Rafailov et al., 2023). These methods are highly effective but can produce models with unexpected biases. One possible explanation, to be tested, is that RLHF preferences contribute to this penalty.

Other alignment strategies include Constitutional AI, which directs models to self-correct based on a set of explicit principles, reducing the need for human labeling of harmful outputs (Bai et al., 2022). Research has also explored methods for directly steering the internal representations of a model to control high-level behaviors (Panickssery et al., 2023). Our work contributes to these efforts by providing a high-resolution diagnostic tool. By measuring a model's sensitivity signature, we can empirically audit the implicit preferences and biases instilled by alignment techniques like RLHF. This allows for a more detailed characterization of how a model interprets instructions, which is a necessary step for developing more predictable and robust control mechanisms.

## 3 Methodology

To quantify the influence of individual adjectives on LLM performance, we adopt a framework based on cooperative game theory. This approach allows us to attribute changes in a model's output accuracy to specific words within a prompt. The methodology consists of a main screening experiment to measure the general impact of 100 adjectives, followed by two focused studies to analyze the effects of prompt structure and word interactions.

### 3.1 Theoretical Framework

We model the contribution of adjectives in a prompt as a cooperative game. In this construction, a set of features, or players, $N = \{1, 2, \ldots, M\}$ cooperate to produce some value. The contribution of each player can be fairly determined using the Shapley value (Shapley, 1953).

#### 3.1.1 Shapley Value Definition

Let $v$ be a value function that maps any subset of players $S \subseteq N$ to a real number, where $v(S)$ represents the total payout generated by the coalition $S$. The Shapley value $\phi_i(v)$ of a player $i \in N$ is the average marginal contribution of player $i$ to all possible coalitions. It is defined as:

$$\phi_i(v) = \sum_{S \subseteq N \setminus i} \frac{|S|!(M - |S| - 1)!}{M!} [v(S \cup i) - v(S)] \tag{1}$$

where $M = |N|$ is the total number of players. The term $[v(S \cup \{i\}) - v(S)]$ represents the marginal contribution of player $i$ to the coalition $S$. The factorial weighting term averages this contribution over all possible permutations of player orderings.

### 3.2 Practical Approximation with KernelSHAP

Calculating exact Shapley values is computationally intractable for a large number of players, as it requires evaluating the value function $v$ for all $2^M$ possible coalitions. For our feature set of $M = 100$ adjectives, this is infeasible. We therefore use KernelSHAP, a model-agnostic approximation method that unifies Shapley values with other feature attribution methods like LIME (Lundberg & Lee, 2017).

KernelSHAP reframes the calculation of Shapley values as a weighted linear regression problem. For a given input instance, we generate a set of simplified inputs, or coalitions $z' \in \{0, 1\}^M$, where $z_i' = 1$ if feature $i$ is present and 0 if it is absent. We then fit a linear model $g(z')$ to approximate the output of the original model for these simplified inputs:

$$g(z') = \phi_0 + \sum_{i=1}^{M} \phi_i z_i' \tag{2}$$

The coefficients $\phi_i$ of this regression are the estimated Shapley values. The key to this method is the specific weighting kernel $\pi_x(z')$, which ensures that the solution to the weighted least squares problem satisfies the properties of Shapley values. The Shapley kernel is defined as:

$$\pi_x(z') = \frac{M - 1}{\binom{M}{|z'|} |z'| (M - |z'|)} \tag{3}$$

where $|z'|$ is the number of present features in the coalition $z'$. The kernel assigns infinite weight to the two full coalitions ($z'$ contains all ones or all zeros) and distributes the remaining weights among the other coalitions based on their size.

### 3.3 Experimental Design

We map the theoretical framework to our experimental setup as follows.

#### 3.3.1 Feature Set and Players

The set of players $N$ consists of $M = 100$ adjectives sampled uniformly at random from a source list of common English adjectives. No curation or filtering was applied to the sample; the adjectives were not selected for their expected influence on model behavior. This unbiased sampling protocol means the set likely includes many low-impact words, which makes the subsequent observation of a long-tail distribution (Section 4.1.1) more informative: even among common adjectives not chosen for their relevance, a small subset exhibits disproportionate influence. The full list of 100 adjectives is provided in the supplementary materials.

#### 3.3.2 The Value Function

The value function $v(S)$ is defined by the performance of a given LLM on a specific task when prompted with the subset of adjectives represented by the coalition $S$.

**Models.** We analyze five distinct LLMs to compare their sensitivity profiles. These are OpenAI's o3 and gpt-4o-mini, Meta's Llama-3-70b-instruct, Microsoft's phi-3-128k-instruct, and DeepSeek's deepseek-r1.

**Task and Dataset.** Our primary benchmark is MMLU (Hendrycks et al., 2021) due to its breadth of subjects and standardized multiple-choice format. For each of the 57 subjects in the dataset, we randomly sample five questions using a fixed seed for reproducibility. To assess cross-benchmark generalizability, we also conduct a pilot study on the ARC-Challenge benchmark (Clark et al., 2018), a collection of grade-school science questions that emphasizes reasoning ability rather than factual recall. We sample 50 ARC-Challenge questions and apply the same Shapley framework to two models from the main experiment: phi-3 and llama-3-70b-instruct.

**Payout Calculation.** For a given question and a coalition of adjectives $S$, we format a prompt containing the question, choices, and the adjectives in $S$. We query the target model and parse its response to extract the chosen letter. The value function is then binary:

$$v(S) = \begin{cases} 1 & \text{if model output is the correct answer} \\ 0 & \text{otherwise} \end{cases} \tag{4}$$

The performance of the model with no adjectival modifiers is defined as the baseline, $v(\emptyset)$. For each question, we estimate the Shapley values by sampling 200 coalitions and fitting the weighted linear model as described in Section 3.2.

### 3.4 Follow-up Study Designs

To investigate the robustness and compositionality of the observed effects, we conducted two follow-up studies on a subset of the most impactful adjectives identified in the main experiment.

#### 3.4.1 Template Variance Analysis

To test if the observed effects are dependent on prompt structure, we re-ran the Shapley analysis on a subset of adjectives using two alternative prompt templates in addition to the original.

- **Original Template** places the adjectival instruction in the preamble.
- **Suffix Template** places the same instruction at the end of the prompt, after the question and choices.
- **Persona Template** reframes the instruction, asking the model to adopt the persona of an expert possessing the adjectival qualities.

By comparing the resulting Shapley values, we can measure the influence of the instruction's syntactic role.

#### 3.4.2 Second-Order Interaction Analysis

To measure how adjectives interact, we calculate the conditional Shapley values. The conditional Shapley value of an adjective $i$, given the fixed presence of another adjective $j$, is denoted as $\phi_i(v_j)$. This is the Shapley value of $i$ in a new game $v_j$ played by the set of players $N \setminus \{j\}$. The value function for this new game is defined for any coalition $S \subseteq N \setminus \{j\}$ as:

$$v_j(S) = v(S \cup j) - v(j) \tag{5}$$

This formulation isolates the marginal contribution of the players in $S$ on top of the baseline performance established by the fixed presence of player $j$. By comparing the conditional value $\phi_i(v_j)$ to the original unconditional value $\phi_i(v)$, we can quantify the interaction effect of adjective $j$ on adjective $i$.

## 4 Results and Analysis

Our empirical investigation reveals a complex landscape of adjectival influence, characterized by consistent structural patterns alongside highly model-specific sensitivities. We find that all models adhere to certain fundamental principles, such as a non-uniform distribution of word impact and a hierarchical sensitivity to task domains. However, the specific responses to these linguistic cues diverge dramatically. The impact rankings vary strongly across model families, indicating that sensitivity patterns are model-specific.

### 4.1 Principles of Adjectival Influence

Despite their architectural diversity, all five models exhibited two consistent behaviors that establish a baseline for understanding linguistic influence.

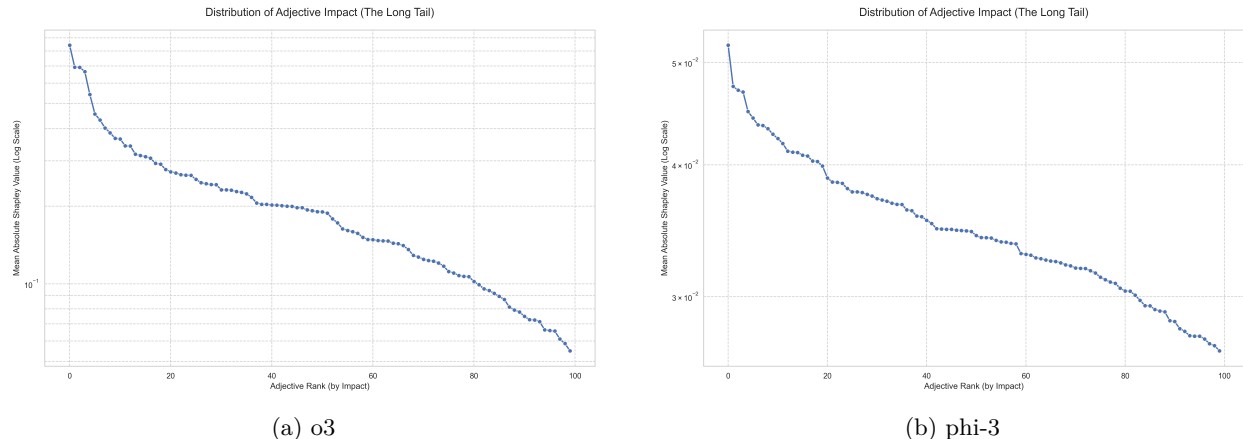

Figure 1: The distribution of adjective impact for o3 and phi-3 , plotted on a log scale. The "long tail" pattern, where a few words have a disproportionately large effect, was observed across all five models, despite the vast difference in the absolute magnitude of their sensitivity.

### 4.1.1 The Long-Tail Distribution of Impact

The influence of adjectives is not evenly distributed. For every model tested, a small subset of adjectives accounts for the vast majority of the measured effect, while most have a negligible influence. This phenomenon is clearly visible in Figure 1, which plots the mean_abs_shapley value for each adjective in descending order for two representative models, o3 and phi-3. The steep initial drop-off, which is best viewed on a logarithmic scale, confirms a "long tail" or Pareto-like distribution of impact. This is a foundational principle of this type of linguistic steering. It indicates that the prompt's semantic space is not flat; rather, certain words act as high-leverage points. Consequently, the task of understanding and controlling model behavior can be made more tractable by focusing on this small set of "power words" rather than the entire vocabulary.

### 4.1.2 Domain Sensitivity

While the overall magnitude of sensitivity varies, the relative ordering of domain types shows a partially consistent pattern. Figure 5 illustrates the average steering magnitude for each model across four broad academic domains. There is considerable variability across the models and no clear pattern emerges in terms of which domains are less susceptible to steering influence.

## 4.2 Cross-Model Divergence

Beyond these general patterns, the models' responses diverge significantly, revealing distinct interpretive frameworks.

### 4.2.1 Uncorrelated Impact Rankings and the Lineage Effect

The specific adjectives that function as "power words" are highly model-dependent. The Spearman rank correlation matrix (Figure 2) quantifies this divergence. A Spearman correlation coefficient ($\rho$) measures the statistical dependence between the rankings of two variables. A value of $+1$ indicates a perfect positive correlation in rankings, while 0 indicates no correlation. The data shows that the correlation of adjective impact rankings between different model families is consistently close to zero. For instance, the correlation between phi-3 and llama-3-70b is only 0.123, and between deepseek-r1 and o3 it is $-0.036$. This indicates that an adjective ranked as highly impactful for one model family offers almost no predictive information about its rank for another.

The only statistically meaningful correlation is between o3 and gpt-4o-mini ($\rho = 0.440$). This "lineage effect" suggests that shared architectural history and training methodologies conserve some sensitivity patterns

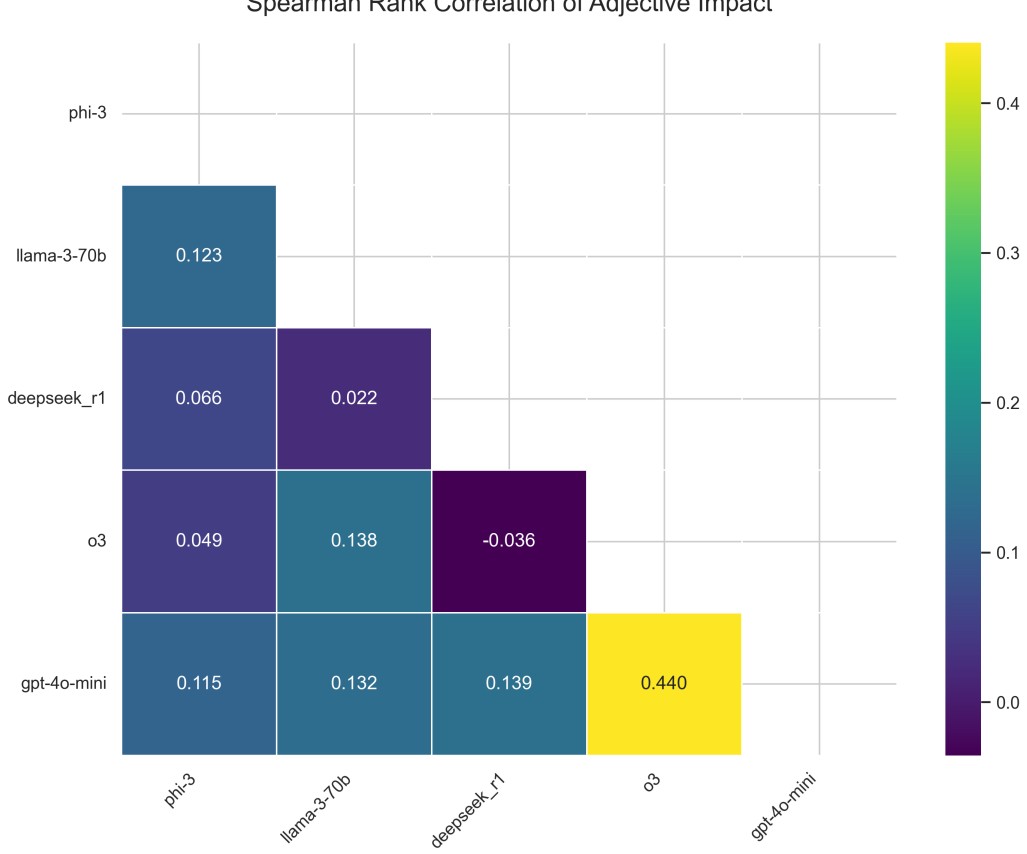

Figure 2: Spearman rank correlation of adjective impact across models. The strong positive correlation between o3 and gpt-4o-mini stands in sharp contrast to the near-zero correlation between all other pairs.

within a specific developer's ecosystem. The lack of broader correlation, however, demonstrates that a universal guide to impactful prompt words is not currently feasible.

### 4.2.2 The Reversal Paradox and Opposing Fine-Tuning Philosophies

The most striking evidence of divergent behavior is the systematic inversion of effects for key adjectival archetypes, a phenomenon we term the Reversal Paradox. Figure 3 compares the directional steering profiles of o3 and gpt-4o-mini, two models from the same developer.

For o3, adjectives related to an "Authority & Formality" archetype, such as academic, are the most performance-enhancing. Conversely, words implying a "Simplicity & Style" archetype, like active and plain, are the most detrimental. gpt-4o-mini exhibits a near-perfect inversion of this preference. academic becomes the single most performance-degrading (negative) adjective, while active and plain become strongly positive. The direction of the effect differs between the two models, suggesting that fine-tuning choices influence sensitivity.

### 4.2.3 A Spectrum of Sensitivity

The models exist on a spectrum of overall sensitivity to linguistic steering. Figure 5 shows that the deepseek-r1 model is a clear outlier, exhibiting a sensitivity an order of magnitude greater than the other models. This is also visible in its steering profile (Figure 4, left), which is far more dispersed than that of a less sensitive model like phi-3. The variance of Shapley values for deepseek-r1 is an order of magnitude higher than for

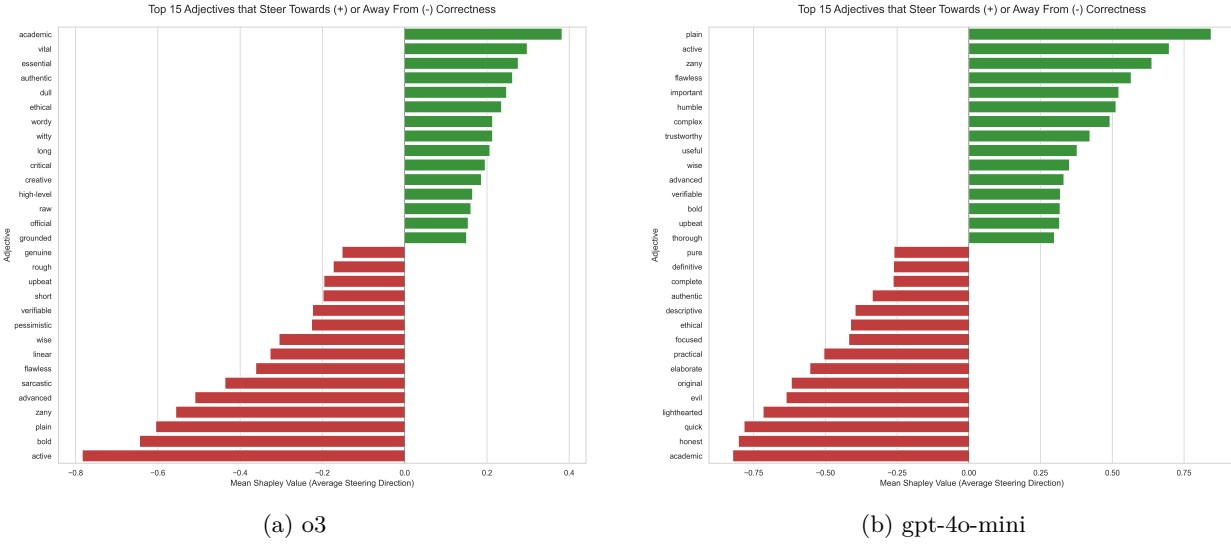

Figure 3: Comparison of directional steering for o3 (left) and gpt-4o-mini (right). The plots show an inversion in their response to key adjectival archetypes. Words that are strongly positive for o3 (e.g., academic) are strongly negative for gpt-4o-mini, and vice versa.

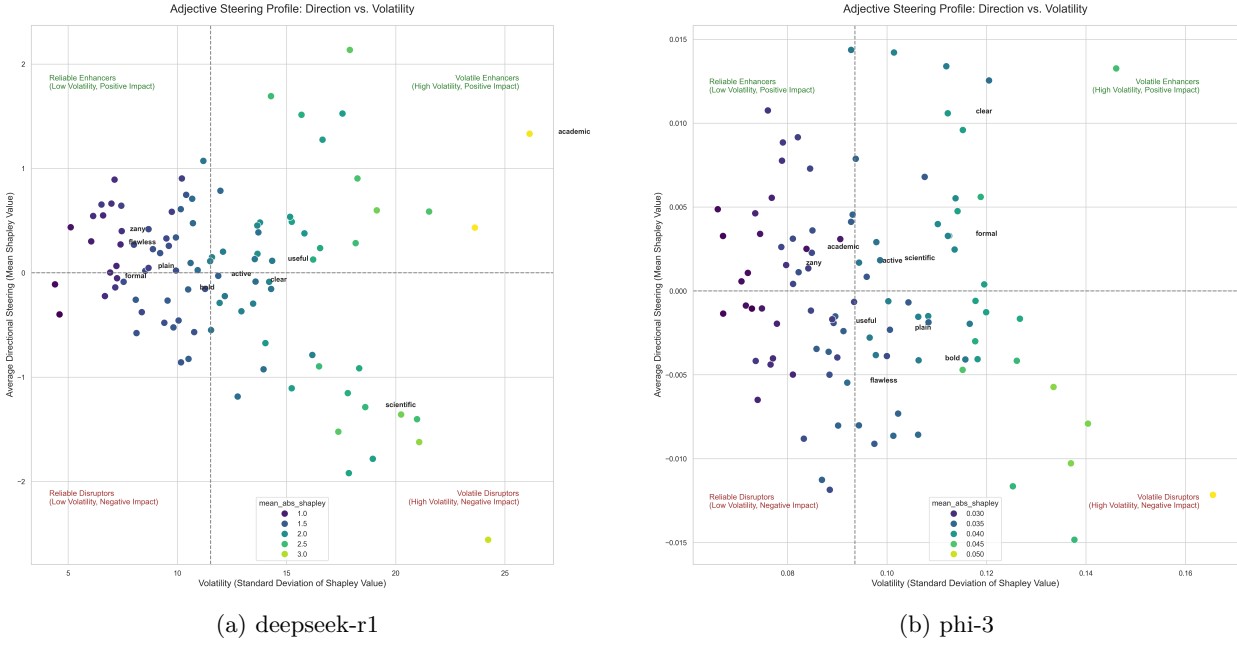

Figure 4: Comparison of Adjective Steering Profiles for the hyper-sensitive deepseek-r1 (left) and the insensitive phi-3 (right). The wide dispersal of points for deepseek-r1 illustrates its high volatility and sensitivity, while the tight clustering for phi-3 shows its robustness to these modifiers.

other models. In contrast, the phi-3 profile is a tight cluster around the origin, indicating that almost no adjectives have a significant or reliable steering effect on this smaller model.

Table 1: Mean Shapley values under three prompt templates for each model's top-5 adjectives. A † marks cases where the persona template reverses the sign of the original effect.

| Model | Adjective | Original | Suffix | Persona |
|---|---|---|---|---|
| gpt-4o-mini | academic | $-0.006$ | $+0.009$ | $+0.003^{\dagger}$ |
| | quick | $-0.003$ | $+0.013$ | $+0.007^{\dagger}$ |
| | active | $-0.006$ | $-0.002$ | $+0.006^{\dagger}$ |
| | sarcastic | $-0.008$ | $-0.028$ | $-0.013$ |
| | bold | $+0.008$ | $-0.001$ | $+0.005$ |
| o3 | active | $+0.001$ | $+0.005$ | $+0.003$ |
| | plain | $-0.001$ | $-0.000$ | $+0.001^{\dagger}$ |
| | bold | $-0.000$ | $-0.001$ | $+0.002^{\dagger}$ |
| | zany | $+0.002$ | $-0.005$ | $-0.001^{\dagger}$ |
| | advanced | $-0.001$ | $+0.000$ | $-0.001$ |
| phi-3 | long | $-0.037$ | $-0.138$ | $+0.008^{\dagger}$ |
| | vague | $-0.073$ | $-0.082$ | $-0.052$ |
| | imaginative | $-0.026$ | $-0.060$ | $+0.013^{\dagger}$ |
| | primary | $+0.018$ | $+0.039$ | $+0.011$ |
| | pessimistic | $-0.052$ | $-0.028$ | $-0.005$ |
| llama-3-70b | linear | $+0.037$ | $+0.061$ | $+0.014$ |
| | sarcastic | $-0.072$ | $-0.255$ | $-0.136$ |
| | descriptive | $-0.043$ | $-0.132$ | $-0.102$ |
| | complex | $-0.024$ | $-0.099$ | $+0.021^{\dagger}$ |
| | advanced | $+0.083$ | $+0.014$ | $+0.027$ |
| deepseek-r1 | accurate | $+0.010$ | $+0.084$ | $+0.003$ |
| | instructive | $-0.067$ | $-0.244$ | $-0.016$ |
| | clear | $+0.014$ | $+0.031$ | $+0.039$ |
| | advanced | $-0.005$ | $-0.039$ | $+0.049^{\dagger}$ |
| | zany | $-0.253$ | $-0.230$ | $-0.118$ |

## 4.3 The Structure of Contextual Influence

Follow-up studies on a subset of the most impactful adjectives reveal that their measured effects are not static but are systematically modulated by prompt structure and composition.

### 4.3.1 Syntactic Roles and the Persona Effect

The impact of an adjective is dependent on its syntactic role. We tested three prompt templates—the original preamble instruction, a suffix placement, and a persona framing ("adopt the persona of an extremely {adjective} expert")—on each model's top-5 adjectives and report the mean Shapley values in Table 1.

Two clear patterns emerge from Table 1. First, the suffix template, which places the adjectival instruction after the question, tends to *amplify* the magnitude of the original effect in both directions. This is particularly pronounced for llama-3-70b (e.g., *sarcastic*: $-0.072 \rightarrow -0.255$) and deepseek-r1 (e.g., *instructive*: $-0.067 \rightarrow -0.244$). Second, the persona template exhibits the opposite tendency: it *attenuates* the magnitude of the original effect and, critically, reverses the sign of the effect in 9 out of 25 adjective–model combinations (marked with †). These reversals occur across all five models, confirming that the persona framing is a distinct and potent modulator of adjectival influence. For example, *academic* shifts from a negative to a positive association for gpt-4o-mini under the persona template, while *advanced* undergoes a large sign reversal for deepseek-r1 ($-0.005 \rightarrow +0.049$). This suggests that the persona instruction activates a qualitatively different interpretive mode rather than simply rescaling the same underlying sensitivity.

### 4.3.2 Compositionality and Conserved Semantic Relationships

Models process adjectives compositionally, not as an independent bag of words. This is evidenced by strong interaction effects where one adjective can amplify or dampen the effect of another. An analysis of the

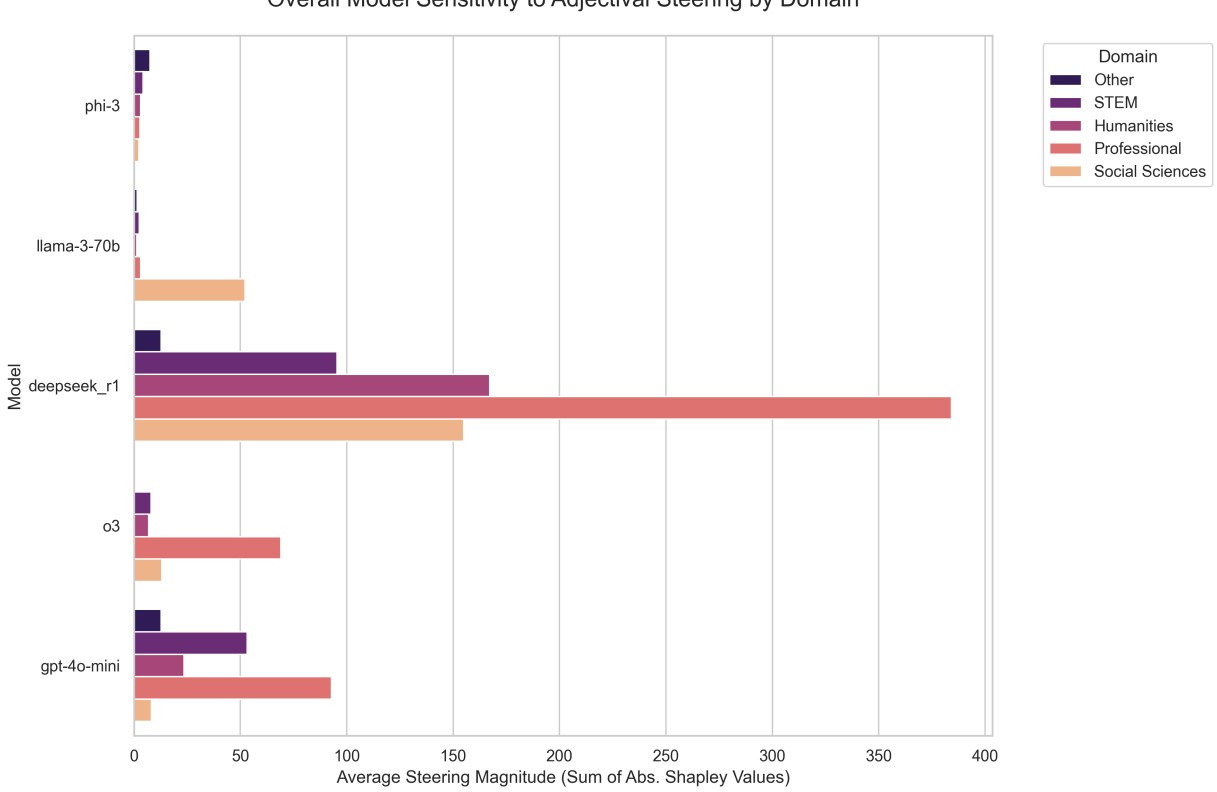

Figure 5: Overall model sensitivity to adjectival steering, broken down by broad academic domain. Note the distinct peak sensitivity for each model family, and the extreme magnitude for deepseek-r1.

correlation heatmaps (e.g., Figure 6) reveals a remarkable finding: even when main effects are inverted between models, the underlying semantic relationships between words can be conserved. For example, in both o3 and gpt-4o-mini, the adjective academic is strongly negatively correlated with active and plain. The models agree that these concepts are in opposition. The difference is that o3 has been trained to value the "academic" pole of this axis, while gpt-4o-mini has been trained to value the "active/plain" pole. This indicates that fine-tuning operates, at least in part, by assigning preference values to pre-existing semantic structures.

## 4.4   Statistical Validation of Adjective Impact

The initial analysis identified a small set of high-impact "power words" for each model based on their mean absolute Shapley values. To formally assess whether these observed effects are statistically significant or merely artifacts of sampling variability, we conducted a formal hypothesis test.

We employed the non-parametric Wilcoxon signed-rank test. This test was chosen because it does not assume that the underlying distribution of per-question Shapley values is normal. Given the potential for outliers and skewed distributions in model performance data, the Wilcoxon test provides a more robust assessment.

For each of the five models, we tested the top 10 most impactful adjectives from our initial ranking. For all 50 tests conducted (10 top adjectives across 5 models), the null hypothesis was rejected with a high degree of confidence ($p < .001$). This provides statistical evidence that the steering effects measured for these key adjectives are significant and non-random.

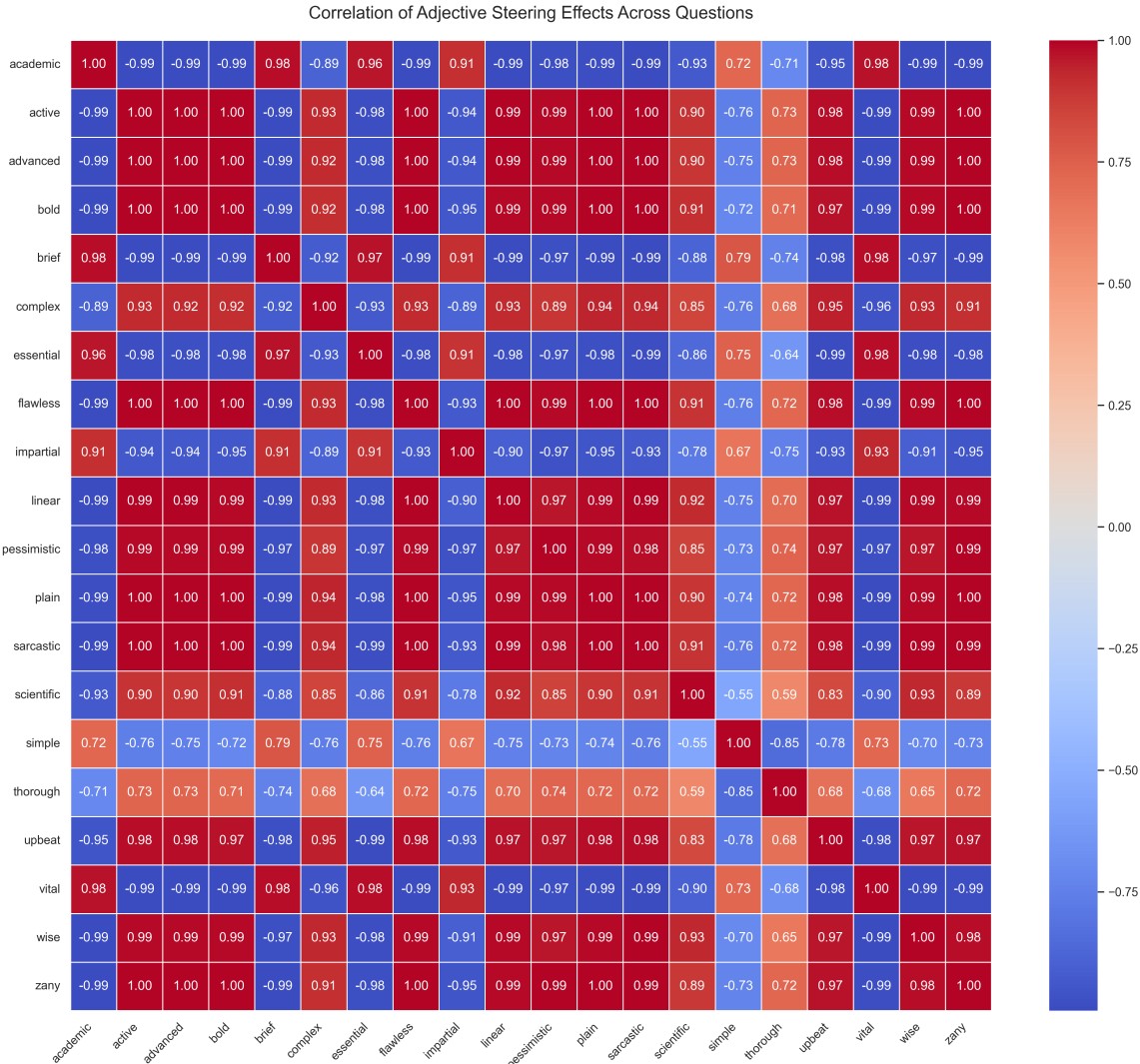

Figure 6: Correlation of adjective steering effects across questions for o3. The strong negative correlation (blue) between academic and words like active, bold, and plain highlights a learned semantic opposition.

## 4.5 Cross-Benchmark Validation

To assess whether our findings generalize beyond the MMLU benchmark, we conducted a pilot study on the ARC-Challenge dataset (Clark et al., 2018) using phi-3 and llama-3-70b-instruct, two models from the main experiment. The results reveal a clear separation between structural properties that generalize and specific rankings that do not.

**The long-tail distribution generalizes.** Table 2 reports the cumulative share of total impact attributable to the top-$K$ adjectives. The concentration of influence is consistent across benchmarks and models: on both MMLU and ARC-Challenge, the top-10 adjectives account for 13–14% of total impact, and the Gini coefficients of impact inequality are comparable (0.089 and 0.127 on ARC vs. 0.084 and 0.390 on MMLU for phi-3 and llama-3-70b respectively). The higher Gini for llama-3-70b on MMLU reflects this model's greater overall sensitivity, but the long-tail shape is preserved across all conditions. This confirms that the long-tail distribution is a structural property of how LLMs respond to adjectival modifiers, independent of the specific task.

Table 2: Cumulative share of total absolute Shapley impact by top-$K$ adjectives, and Gini coefficient of impact inequality. The long-tail concentration is consistent across benchmarks despite different adjective rankings.

| Benchmark | Model | Top-5 | Top-10 | Top-20 | Gini |
|---|---|---|---|---|---|
| MMLU | phi-3 | 6.8% | 13.0% | 24.7% | 0.084 |
| MMLU | llama-3-70b | 16.2% | 28.9% | 45.7% | 0.390 |
| ARC-Challenge | phi-3 | 6.8% | 13.1% | 24.8% | 0.089 |
| ARC-Challenge | llama-3-70b | 7.7% | 14.2% | 26.6% | 0.127 |

**Adjective rankings are benchmark-specific.** Despite the structural similarity, the Spearman rank correlation of adjective impact between MMLU and ARC-Challenge for the *same* model is indistinguishable from zero: $\rho = 0.002$ ($p = 0.98$) for phi-3 and $\rho = 0.019$ ($p = 0.85$) for llama-3-70b. The overlap in top-10 adjectives between benchmarks is minimal (1 out of 10 for both models). Cross-model correlations within ARC are similarly near zero ($\rho = -0.013$ between phi-3 and llama-3-70b). This indicates that while the existence of high-impact "power words" is a universal phenomenon, the identity of those words is contingent on both the model and the task, reinforcing the need for empirical characterization of each model–task combination.

**A faint lineage signal persists.** An additional ARC-Challenge run with llama-3-8b-instruct allows us to test whether the lineage effect observed on MMLU (Section 2) extends across model scales. Within ARC, the Spearman correlation between llama-3-8b and llama-3-70b is $\rho = 0.182$ ($p = 0.069$), while both Llama variants correlate near zero with phi-3 ($\rho = 0.061$ and $\rho = -0.013$). The within-family correlation is modest compared to the $\rho = 0.440$ observed between o3 and gpt-4o-mini on MMLU, consistent with the much larger capacity gap between 8b and 70b. This suggests that the lineage effect scales with model similarity within a family, not merely with shared developer origin.

## 5 Discussion

The results of our comparative analysis provide a detailed view of the complex relationship between linguistic inputs and LLM behavior. The findings suggest that the association between adjectival modifiers and model performance is not a simple function of word meaning but is instead governed by a combination of factors. We discuss the implications of these findings for understanding model behavior and for the broader field of AI alignment.

### 5.1 Implications for AI Alignment and Controllability

Our findings have several direct consequences for the study and practice of AI alignment.

First, the lack of a universal steering vocabulary across different model families indicates that alignment techniques targeting specific linguistic cues may not be broadly transferable. The "lineage effect" we observe, where only models from the same developer show correlated sensitivities, suggests that fine-tuning details and training data create idiosyncratic sets of control levers. An alignment intervention that relies on a word like honest being a net negative for gpt-4o-mini could fail or even have an unintended effect when applied to a model like o3, for which the word is a net positive.

Second, the "persona effect" represents a promising direction for robust control. The finding that models respond differently and often more favorably to a persona instruction ("act as an X expert") than a direct command ("be X") suggests that the persona framing is associated with more favorable outcomes than direct adjectival commands. This may be because personas provide a coherent set of desirable attributes.

Third, the existence of model-specific "domain sensitivity signatures" provides a method for auditing and characterizing model behavior. By identifying the domains where a model is most susceptible to linguistic steering, we can identify areas where it is most likely to exhibit unpredictable behavior and where alignment interventions are most needed. The extreme sensitivity of deepseek-r1 in professional domains, for example,

suggests that its application in high-stakes fields would require extensive testing for robustness against prompt variations.

## 5.2 Computational Cost and Practical Feasibility

The KernelSHAP framework requires a substantial number of model queries, and the cost–precision tradeoff merits explicit discussion. For the main screening experiment, each of the 285 questions was evaluated with 200 sampled coalitions, producing $285 \times 200 = 57{,}000$ inference calls per model and 285,000 calls across all five models. The follow-up template variance study required an additional $285 \times 100 \times 3 = 85{,}500$ calls per model (100 coalitions across three templates), and the interaction study required $285 \times 100 \times 5 = 142{,}500$ calls per model (100 coalitions for each of five fixed-context conditions). The total experimental cost was therefore approximately 1,425,000 inference calls.

The choice of 200 coalitions for $M = 100$ features in the main experiment represents a practical compromise. While full enumeration of $2^{100}$ coalitions is infeasible, the 200-sample KernelSHAP approximation provides sufficient precision to detect the large-effect adjectives that drive our primary conclusions, as validated by the Wilcoxon signed-rank tests ($p < .001$ for all top-10 adjectives across all models; see Section 4.4). Mid-range adjectives are estimated with greater noise, which is reflected in their higher standard deviations. The follow-up studies use only 100 coalitions but with just $M = 5$ features, a ratio that provides considerably tighter estimates. For practitioners, these costs represent a one-time characterization per model, and the resulting sensitivity profiles can be reused to inform prompt design across applications.

## 5.3 Limitations

This study has several limitations that define the scope of its conclusions. First, while a pilot study on ARC-Challenge confirms that the long-tail distribution generalizes across benchmarks, our primary analysis is focused on multiple-choice question-answering tasks. The observed effects may not generalize to other formats, such as creative writing or open-ended dialogue, where the utility of certain adjectives could be different. Second, our feature set was limited to 100 adjectives. Other parts of speech, such as adverbs or specific nouns, may also function as powerful steering agents. Third, our use of KernelSHAP provides an approximation of true Shapley values, and the analysis is correlational, identifying patterns of influence without fully explaining the underlying causal mechanisms within the models. Accordingly, throughout this paper, terms such as "effect" and "influence" refer to statistical associations measured via Shapley value attribution, not to established causal mechanisms.

## 5.4 Future Work

The limitations of this study suggest several avenues for future research. Our ARC-Challenge pilot suggests that structural properties generalize while specific rankings do not; extending this analysis to open-ended generation, dialogue, and creative tasks would determine the full scope of this finding. A second direction would be to expand the feature set to other word types to build a more complete grammar of linguistic influence. Finally, connecting these black-box, observational findings to the internal workings of models through mechanistic interpretability techniques could provide a causal explanation for why certain words activate specific behavioral modes, forming a bridge between empirical prompt engineering and a formal science of model control.

## 6 Conclusion

This work provides a systematic, quantitative analysis of how individual adjectives are associated with changes in the behavior of a diverse set of Large Language Models. We move beyond anecdotal evidence to show that a small number of "power words" possess a disproportionately large influence on model performance. Our central finding is that the effect of these words is not universal but is instead a function of a model's specific style and training lineage. We identified a Reversal Paradox between models from the same developer and found that each model family exhibits a unique domain sensitivity signature. Furthermore,

we demonstrated that the impact of any given adjective is highly contextual, modulated by its syntactic role and compositional interactions with other words.

The results show that while the existence of powerful linguistic levers appears to be a consistent property of LLMs, the manual for operating these levers is different for each model. A robust and predictable science of AI alignment will therefore require an empirical, model-specific approach, beginning with the characterization of a model's unique response patterns to build safe and effective control mechanisms.

## Code Availability

All code and data required to reproduce the experiments are publicly available at `https://github.com/lmlearning/linguisticsteering`. The repository includes the main Shapley estimation scripts for both MMLU and ARC-Challenge, the analysis and visualization code, cross-benchmark analysis tools, and the adjective lists used in this study. Raw result files with per-question Shapley values are included in the supplementary materials.

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

# A   Reproducibility Guide

This appendix serves as a guide for reproducing the linguistic steering experiments. It details the environment setup, main experiment execution, data post-processing, and analysis phases required to replicate the findings.

## A.1   Step 1: Environment Setup

Before running any scripts, you must configure your environment and install the necessary Python packages. This project relies on several data science libraries and specific API clients for LLMs.

1. **Install Python Packages:** The project does not include a single setup script. You must install the required packages manually, likely using a requirements.txt file or pip. Based on the imports in the scripts, key dependencies include:

   - Data handling: `pandas`, `numpy`
   - Datasets: `datasets` (from Hugging Face)
   - Machine Learning: `scikit-learn`
   - Plotting: `matplotlib`, `seaborn`
   - API Clients: `openai`, `replicate`, `google-generativeai`

2. **Set API Keys:** The core script requires API credentials for the LLM provider you intend to use. You must provide these either as command-line arguments or by setting them as environment variables.

   - `OPENAI_API_KEY`
   - `REPLICATE_API_TOKEN`
   - `GOOGLE_API_KEY`

3. **Adjective Lists:** Ensure the adjective files (`adjectives08.txt`, `adjectives100.txt`) are present in the project directory.

## A.2 Step 2: Phase 1 - Main Experiment Execution

The primary data generation is handled by `estimate_importance.py`. This script samples questions from the MMLU dataset, queries an LLM with prompts modified by different combinations of adjectives, and uses a regression model to estimate the Shapley value or influence of each adjective on the model's correctness.

To run the main experiment, execute the following command, customizing the parameters for your chosen setup:

```
# Example using OpenAI's gpt-4o with the 100-adjective list
python estimate_importance.py \
    --adjectives_file adjectives100.txt \
    --api_provider openai \
    --model gpt-4o \
    --openai_api_key "YOUR_API_KEY_HERE" \
    --output_file "gpt4o_results.json" \
    --num_per_category 5 \
    --num_samples 200 \
    --max_concurrent_requests 10
```

This process can be lengthy and expensive, as it makes many API calls. The script is designed to be resumable; if it is interrupted, it will load the existing output file and continue where it left off.

## A.3 Step 3: Phase 2 - Data Post-Processing (Optional)

The script `correct_outlier.py` is a utility for data cleaning. If the initial data generation resulted in numerical artifacts or extreme outlier values (as noted in the script's default threshold of $10^3$), this script can be used to normalize them.

```
# Example of correcting the raw results and saving to a new file
python correct_outlier.py \
    --input_file gpt4o_results.json \
    --output_file gpt4o_corrected.json \
    --value 1e9 \
    --threshold 1e3
```

### A.4   Step 4: Phase 3 - Primary Analysis and Visualization

Once you have a clean results file (e.g., `gpt4o_corrected.json`), you can generate the main analysis report using `analyze.py`. This is the core analysis script that produces most of the key tables and plots discussed in the paper.

```
# This script takes a single JSON file as input
python analyze.py gpt4o_corrected.json
```

This command will create a new directory named `gpt4o_corrected_analysis/`. Inside, you will find:

- `_summary.json`: A JSON file with key statistics, like top adjectives and domain sensitivity.

- `_summary_table.md`: A Markdown file ranking all adjectives by their overall impact.

- Multiple plots (`.png` files), including the mean Shapley values, overall impact distribution, and the quadrant analysis.

### A.5   Step 5: Phase 4 - Secondary and Meta-Analyses

The remaining scripts perform more targeted analyses on the generated JSON results.

**Variance and Hypothesis Testing.**   To assess the statistical significance and stability of the results, use `analyze_variance.py` and `onesample_test.py`.

```
# Perform bootstrap analysis on the top 10 adjectives
python analyze_variance.py \
    --input_results gpt4o_corrected.json \
    --top_k 10

# Run a one-sample t-test to check if impact is > 0
python onesample_test.py \
    --input_results gpt4o_corrected.json \
    --top_k 10
```

**Comparative and Meta-Analysis.**   If you have run the experiment with multiple models, you can compare their results using `correlation_ranks.py` and `meta_analysis.py`.

```
# Calculate the rank correlation between two result files
python correlation_ranks.py \
    gpt4o_corrected.json \
    o3_corrected.json \
    --output ranks.csv

# Run a full meta-analysis across multiple analysis directories
python meta_analysis.py \
    GPT4:gpt4o_corrected_analysis/ \
    Opus:o3_corrected_analysis/
```

**Follow-up Experiments.**   The `follow_up_analysis.py` script performs more advanced experiments (template variance, second-order interactions) based on the initial results. It re-runs a smaller set of API calls.

```
# Run follow-up tests on the top 5 adjectives from the first run
python follow_up_analysis.py \
    --input_results gpt4o_corrected.json \
    --top_k 5
```

**ARC-Challenge Cross-Benchmark Pilot.**  The `estimate_importance_arc.py` script runs the Shapley analysis on the ARC-Challenge benchmark instead of MMLU. It supports local inference via any OpenAI-compatible endpoint (e.g., `llama-server` from llama.cpp). The ARC pilot experiments used Q4_K_M quantized GGUF models, seed 42, 50 questions, and 200 coalitions.

```
# Start a local llama-server (example with phi-3)
llama-server -m Phi-3-mini-128k-instruct.Q4_K_M.gguf \
    --port 8081 -ngl auto

# Run the ARC pilot
python estimate_importance_arc.py \
    --adjectives_file adjectives100.txt \
    --base_url http://localhost:8081/v1 \
    --output_file arc_phi3_results.json \
    --num_questions 50 --num_samples 200 --seed 42

# Cross-benchmark analysis
python analyze_cross_benchmark.py \
    --arc arc_phi3_results.json arc_llama8b_results.json \
    --mmlu shap_results_replicate.json results_llama.json \
    --labels phi-3 llama-3-8b phi-3 llama-3-70b
```

