# OpenReview forum: "Investigating Linguistic Steering: An Analysis of Adjectival Effects Across Large Language Model Architectures"
_TMLR — Accepted by TMLR_

### Review · Reviewer_iPTK · 2025-12-12

**Summary Of Contributions:**

This work uses Shapley values to evaluate how individual adjectives in prompts affect LLM output. The authors apply KernelSHAP to 100 adjectives across five models (o3, gpt-4o-mini, phi-3, llama-3-70b, deepseek-r1) on the MMLU benchmark.

Key strengths:
1. The proposed Shapley value-based method provides a theoretical framework for prompt optimization.
2. The finding of "lineage effect" that models from the same developer show correlation on adjective sensitivities, while cross-family models show no correlations.
3. Detailed document and code in the appendix for reproducing the results.

**Audience:**

Yes

**Audience Explanation:**

Yes. The paper investigates the role of adjectives in prompts, which is relevant to LLM researchers and prompt engineering practitioners.
The work provides findings and observations that steering effects don't transfer across model families. The used Shapley value-based method may inspire the interpretability and attribution methods community.

**Broader Impact Concerns:**

No concerns

**Claims And Evidence:**

No

**Claims Explanation:**

While the paper proposes an interesting Shapley-value framework for analyzing adjectival steering effects, the following weaknesses make the claims not fully supported:
1. The sample size is too small. Only 5 questions per MMLU subject (285 total) is too small to reliably estimate Shapley values for 100 adjectives. The authors should conduct larger-scale experiments to validate the conclusions.
2. All the experiments are conducted on the MMLU task. The conclusion may not generalize to other tasks, such as generation. The experiments on other tasks or datasets are suggested.
3. The paper claims persona prompts can invert adjective effects in section 4.3.1. But no quantitative evaluation results are provided to support the claim.

**Requested Changes:**

Please refer to the weaknesses.

---

### Review · Reviewer_DckK · 2025-12-23

**Summary Of Contributions:**

This paper presents a systematic, model-agnostic investigation into how individual adjectives influence the behavior of large language models (LLMs) across different architectures. The core contribution is a principled framework based on Shapley values—a game-theoretic method—to quantify the “steering effect” of linguistic cues in prompts. Rather than treating prompts as black-box inputs, the authors decompose their semantic components and measure how each adjective independently or jointly shifts model performance on the MMLU benchmark. By applying this approach to five diverse LLMs—including models from OpenAI, Meta, Microsoft, and DeepSeek—the study reveals several important patterns: (1) only a small subset of adjectives consistently exert strong influence, (2) models from the same developer lineage show correlated sensitivity (“lineage effect”), while others exhibit near-zero correlation in their responses, and (3) syntactic context and compositional interactions dramatically modulate adjective effects, sometimes even reversing their direction. Notably, the paper identifies a “Reversal Paradox,” where two models from the same family respond oppositely to the same adjectives (e.g., academic helps one but harms the other), highlighting how fine-tuning choices create divergent alignment behaviors. A key strength is the rigorous, reproducible methodology grounded in established interpretability tools (KernelSHAP) and validated with statistical testing. The work also includes a detailed reproducibility guide, enhancing its scientific value. A limitation is its restriction to adjectives and the MMLU task, which may not generalize to other linguistic categories or open-ended settings. Still, the paper convincingly demonstrates that linguistic steering is highly model-specific and context-dependent—challenging one-size-fits-all prompt engineering and underscoring the need for empirical, model-aware alignment techniques.

**Audience:**

Yes

**Audience Explanation:**

The paper directly addresses core TMLR themes such as understanding how intelligent systems interpret linguistic cues, the empirical behavior of learning systems under controlled interventions, and the development of analytical frameworks for model interpretability

**Claims And Evidence:**

Yes

**Claims Explanation:**

The paper presents a methodologically rigorous framework grounded in Shapley values from cooperative game theory to quantify how individual adjectives influence the behavior of large language models (LLMs). The authors evaluate 100 adjectives across five diverse LLMs (including o3, gpt-4o-mini, and Llama-3-70b) on the MMLU benchmark, using KernelSHAP—a well-established approximation technique—for scalable attribution. The evidence is both quantitatively robust and statistically validated

**Requested Changes:**

Clarify the causal interpretation of Shapley values: While the paper correctly uses Shapley values as a feature attribution tool, it occasionally implies causal language (e.g., “adjectives steer model behavior”). To avoid overinterpretation, the authors should explicitly note that this is an observational, correlational analysis—Shapley values here reflect statistical contributions, not causal interventions.

Expand discussion on computational cost and scalability (strengthening): The paper mentions using 200 sampled coalitions but doesn’t discuss the trade-off between approximation accuracy and computational expense, especially for API-based models. A brief note on feasibility for wider adoption would help practitioners.

---

### Review · Reviewer_7G8b · 2026-01-18

**Summary Of Contributions:**

This paper introduces a game-theoretic framework using Shapley values (approximated via KernelSHAP) to quantify the steering effect of 100 adjectives on the performance of five Large Language Models (LLMs) including o3, gpt-4o-mini, and deepseek-r1. The authors conduct experiments on the MMLU benchmark to attribute changes in model accuracy to specific adjectival modifiers. Key findings include a "long-tail" distribution of adjectival impact, a "lineage effect" where models from the same developer exhibit correlated sensitivities, and a "reversal paradox" where semantic levers work in opposite directions for different models. The study also explores the influence of syntactic roles (e.g., personas) and compositional interactions between adjectives.

**Audience:**

Yes

**Audience Explanation:**

At least some individuals in TMLR’s audience would be interested in the findings of this paper because it directly addresses a core scientific question relevant to the machine learning research community: how linguistic inputs systematically influence the behavior of large language models, and how this influence varies across model architectures and training lineages.

From a methodological perspective, the paper introduces a principled, model-agnostic framework based on Shapley values to attribute performance changes to individual prompt components. This goes beyond heuristic prompt engineering and contributes to the broader literature on feature attribution, interpretability, and black-box analysis of foundation models, which are central topics within TMLR’s scope.

From a scientific insight perspective, the findings reveal nontrivial and previously underexplored phenomena, including:
1. The long-tail distribution of linguistic influence, where only a small subset of words act as high-leverage control signals.
2. The lineage effect, showing that prompt sensitivity profiles correlate within model families but diverge sharply across architectures.
3. The reversal paradox, where the same linguistic modifier can have opposite effects across closely related models.
4. Evidence of non-additive, compositional interactions between prompt elements in larger models.

**Broader Impact Concerns:**

No concerns

**Claims And Evidence:**

Yes

**Claims Explanation:**

The paper presents a timely and empirically grounded investigation into the mechanics of prompt engineering, moving the field from anecdotal heuristics to more rigorous attribution.

1. Originality & Significance: The identification of the "Lineage Effect" and "Reversal Paradox" (visualized effectively in Figure 3) provides a significant contribution to our understanding of how fine-tuning shapes model behavior. The finding that sensitivity profiles are developer-specific rather than universal challenges the current "one-prompt-fits-all" paradigm.
2. Methodological Framework: The application of Shapley values to prompt engineering is a principled choice, offering a structured way to decompose the "black box" of in-context learning.
3. Clarity: The paper is well-written and the visualization of results,  figure 2 effectively communicates the lack of transferability in prompting strategies across different model families.

**Requested Changes:**

Overall, the paper is good. I have several concerns:
1. Limited Evaluation Scope: The analysis is restricted solely to the MMLU benchmark. Adjectival steering is highly context-dependent; an adjective like "creative" might be detrimental to multiple-choice QA (MMLU) but highly effective in open-ended generation or reasoning tasks. Generalizing "sensitivity signatures" (as shown in Figure 5) from a single benchmark to general model behavior is a stretch. The "Reversal Paradox" might essentially be a "Task Mismatch Paradox" where certain adjectives shift the model into a mode incompatible with MMLU's rigid format.
2. Undefined Selection of Adjectives: The set of 100 adjectives is described simply as being loaded from a "predefined text file" (Section 3.3.1). There is no justification for how these specific words were chosen. Are they high-frequency words? Randomly sampled? Selected from prompt engineering guidebooks? Without this context, it is impossible to know if the "long tail" distribution (Figure 1) is a fundamental property of language models or a result of selecting 90 irrelevant words and 10 relevant ones.

---

### Author Response · Authors · 2026-01-22
**Response to reviewers**

We thank the reviewers for their careful and diligent engagement with our work and their helpful suggestions. We address each reviewer's concerns below.


### Response to Reviewer 7G8b:

**On adjective selection:** We acknowledge that we should have been explicit about the source of the 100 adjectives and the sampling protocol used. The adjectives were sampled uniformly from a source list containing common adjectives in English. There was no curation of the source list. We acknowledge this means our sample may include low-impact words, but we argue that this makes the long-tail finding more informative as even among common adjectives not selected for their steering potential, a small subset exhibits disproportionate influence. We'll clarify the methodology and discuss its implications in the revision.

**On scope:** We acknowledge that only testing on MMLU is a limitation in our approach and that, obviously, words can have different effects in different generation contexts. We're happy to explicitly reduce the scope of our claims to similar tasks and state this as a limitation, and/or conduct a small-scale pilot on a second benchmark to test the approach further.


### Response to Reviewer DckK:

**On causal language:** We acknowledge that we in places use inappropriately causal language and commit to revising this.

**On computational cost:** We think this is a good suggestion add will add a discussion of the cost-precision tradeoff and practical feasibility.


### Response to Reviewer iPTK:

**On sample size:** We understand the concern, but we believe the statistical picture is strong enough to warrant attention. Many of our key claims are really about patterns across models rather than precise point estimates for individual adjectives. With 285 questions × 5 models × 200 coalition samples, we have substantial power to detect these cross-model patterns. Even on individual word effects, some effects are strong enough to pass statistical muster. We are, however, willing to add additional tests if that would be helpful.

**On scope:** As we noted above in our response to reviewer 7G8b, we acknowledge this limitation and are happy to tone down our claims or add a smaller secondary experiment.

**On the persona effect:** We accept this criticism and will add the explicit quantitative support in the revision. This is an oversight on our part.

---

### Decision · Action_Editor_9N9A · 2026-03-15

**Recommendation:** Accept with minor revision

**Additional Comments:**

As mentioned above, while the reviewers appreciated the proposed Shapley based framework to quantify the steering effect of adjectives,  they shared concerns about the evaluation with the initial submission. The authors did not fix the issues in rebuttal but have committed to resolving them, and all reviewers recommended acceptance conditioned on authors addressing the issues with new experiments and revised presentations as promised. Specifically, the authors should:

1. Performing experiments on new benchmark beside MMLU;

2. Reporting quantitative analysis of  cost-precision tradeoff and the persona effect;

3. Revising the claims of causal language to avoid overinterpretation and clarifying adjective selection.

**Audience:**

Yes

**Audience Explanation:**

The paper investigates the role of adjectives in prompts, which is relevant to LLM researchers and practitioners. The used Shapley value-based method may inspire researcher working on interpretability and alignment. Overall, this paper would be interesting to TMLR audience.

**Claims And Evidence:**

Yes

**Claims Explanation:**

The reviewers unanimously agreed that applying Shapley values to quantify steering effect is an intuitive solution and the method is supported by empirical validations. While the reviewers shared concerns on evaluation with initial submission, such as limited to MMLU benchmark and lacking quantitative analysis of  cost-precision tradeoff and the persona effect, the authors have committed to resolving these issues in their revision.